# Genetic Dissection of Grain Yield and Agronomic Traits in Maize under Optimum and Low-Nitrogen Stressed Environments

**DOI:** 10.3390/ijms21020543

**Published:** 2020-01-15

**Authors:** Berhanu Tadesse Ertiro, Michael Olsen, Biswanath Das, Manje Gowda, Maryke Labuschagne

**Affiliations:** 1Ethiopian Institute of Agricultural Research, Bako National Maize Research Center, P.O. Box 2003, Addis Ababa, Ethiopia; berhanu.te@gmail.com; 2Department of Plant Sciences, University of the Free State, P.O. Box 339, Bloemfontein 9300, South Africa; 3International Maize and Wheat Improvement Center (CIMMYT), P.O. Box 1041, Nairobi 00621, Kenya; M.Olsen@cgiar.org (M.O.); b.das@cgiar.org (B.D.); M.Gowda@cgiar.org (M.G.)

**Keywords:** low nitrogen, maize, marker assisted selection, QTL

## Abstract

Understanding the genetic basis of maize grain yield and other traits under low-nitrogen (N) stressed environments could improve selection efficiency. In this study, five doubled haploid (DH) populations were evaluated under optimum and N-stressed conditions, during the main rainy season and off-season in Kenya and Rwanda, from 2014 to 2015. Identifying the genomic regions associated with grain yield (GY), anthesis date (AD), anthesis-silking interval (ASI), plant height (PH), ear height (EH), ear position (EPO), and leaf senescence (SEN) under optimum and N-stressed environments could facilitate the use of marker-assisted selection to develop N-use-efficient maize varieties. DH lines were genotyped with genotyping by sequencing. A total of 13, 43, 13, 25, 30, 21, and 10 QTL were identified for GY, AD ASI, PH, EH, EPO, and SEN, respectively. For GY, PH, EH, and SEN, the highest number of QTL was found under low-N environments. No common QTL between optimum and low-N stressed conditions were identified for GY and ASI. For secondary traits, there were some common QTL for optimum and low-N conditions. Most QTL conferring tolerance to N stress was on a different chromosome position under optimum conditions.

## 1. Introduction

In Sub-Saharan Africa, most maize is produced under N-deficient conditions, owing to limited availability of resources, low purchasing power of farmers, and low incentive from governments [1,2]. In this scenario, developing cultivars tolerant to low-N stressed environments is crucial for sustainable production and ensuring food security in the region. Contrary to farmers’ practice, most breeding programs in the region develop new varieties under optimally managed on-station experimental plots. The genetic mechanism for grain yield (GY) under optimum and low-N stressed conditions are different, and varieties developed for optimal environments often respond differently under N-limiting environments [2,3]. Understanding the genetic architecture of GY and traits correlated with it would accelerate genetic improvement in maize yield.

GY is the most economically important trait in maize breeding programs in developing countries. Other agronomically relevant traits, including ASI, PH, EH, EPO, and SEN, are often used by breeders to find desirable plant architecture and for indirect selection of high-yielding maize varieties. The availability of reliable large-effect quantitative trait loci (QTL) for GY and other traits under optimum, as well as low-N stressed, conditions would accelerate the development and release of new maize varieties meeting yield demand under optimum and poor soil conditions, particularly for resource-poor farmers. Unfortunately, not much is known about the genetic architecture of most of these traits under N-stressed conditions, and QTL with a major effect have not yet been reported. Breeding for low-N stressed conditions thus far focused on direct selection for GY and indirect selection for correlated secondary traits under N-stressed conditions. Selection based on phenotypic traits is less accurate and expensive compared to marker-based selection.

QTL analysis based on high-density linkage maps will provide the basic understanding of the genetic architecture of quantitative traits, thereby relating specific genetic loci with the biological mechanisms associated with desirable phenotypes [4]. The identification and characterization of QTL will help the breeders/geneticists to recognize genomic regions associated with the expression of complex traits and their precise genetic contribution at target loci. Several QTL studies have been undertaken in an effort to understand the genetic basis of abiotic stress tolerance in maize [5,6,7,8,9,10,11]. However, most studies focused on drought stress, and little research has been done on the dissection of the genetic basis of low-N tolerance. The few studies conducted to understand the NUE and associated traits in maize have given good insight into the genetic basis of low-N tolerance in maize [4,12]. One of the challenges in translating QTL identified into MAS, has been the environment-dependent and genotype-specific nature of QTL identified [13]. Most QTL reported under low and optimum N so far are mainly based on studies from one population in one/few optimum and low-N environments. For example, one mapping population with 240 F_2:3_ families was evaluated under one optimum and two low-N sites in Mexico [12]. Another study [14] evaluated 214 F_3_ families in one location over two seasons. Previous QTL mapping efforts for low-N were conducted in single optimum or low-N stressed sites, using only one mapping population. Multilocation trial data from more than one mapping population would provide a clear picture on the stability of QTL across environments and genetic backgrounds. In this study, five doubled haploid (DH) populations were evaluated in three to five optimum environments and one to three environments under managed low-N stress in the wet and off-seasons. The main objectives of this study were (1) to identify the QTL associated with GY, and other related traits under optimum and low-N stressed (wet and off-season) conditions, and (2) to identify common genomic regions across management conditions, traits, and genetic backgrounds. The identification of major QTL for GY and/or other traits that are common across different N conditions and genetic backgrounds would facilitate the application of MAS for the improvement of grain yield under low-N stress conditions.

## 2. Results

### 2.1. Trial Mean, Genetic Variance, and Heritability of Traits

Increase in the intensity of N stress decreased trial mean for GY, PH, EH, and EPO, and increased trial mean for ASI and SEN (Figure 1). Average genetic variance in all populations was higher under optimum conditions for GY, PH, EH, and EPO, but it was high under low-N conditions for AD, SEN, and ASI. Despite relatively higher genetic variance under optimum than low-N conditions, broad sense heritability for GY and most secondary traits under low-N and optimum conditions were on par. Phenotypic and genetic correlations for GY were consistently positive and significant with PH, EH, EPO, and AD.

### 2.2. QTL Mapping in Five DH Populations

A total map length of 3688.3, 4004.6, 3871.9, 7193.1, and 3426.4 cM was obtained from 2104, 2699, 1962, 1985, and 2086 SNP markers (Table 1) for populations 1, 2, 3, 4, and 5, respectively. The average distance between adjacent markers ranged from 1.48 cM for population 2 to 3.62 cM for population 4.

QTL analysis identified 155 significant QTL for GY, AD, ASI, PH, EH, EPO, and SEN across ten maize chromosomes under optimum (55), LNM (49), and LNO (51) conditions (Table 2). Though slightly higher under optimum conditions, the total number of QTL identified for all N conditions, traits, and populations were comparable. The total number of QTL identified for GY, AD, ASI, PH, EH, EPO, and SEN were 13, 43, 13, 25, 30, 21, and 10, respectively. The distribution of QTL was variable among chromosomes, ranging between 6 (chromosome 9) and 51 (chromosome 1), with an average of 15.5 QTL in each chromosome. The three chromosomes with the largest number of QTL were chromosome 1 (51), chromosome 3 (26), and chromosome 8 (20). The distribution of the QTL across the five populations was 28, 84, 16, 13, and 14 for population 1, 2, 3, 4, and 5, respectively.

For GY, 13 significant QTL were identified under optimum (3), LNM (2), and LNO (8) conditions across all chromosomes, except chromosomes 5, 6, and 9 (Table 3). Common QTL for optimum and low-N stressed conditions were not identified in all five populations. QTL underlying GY under optimum, LNM, and LNO conditions were identified on chromosomes 1, 2, 7, and 10 of population 2. About 62% of all QTL for GY individually contributed more than 10% of the observed phenotypic variance. The proportion of phenotypic variance explained by each QTL varied between 6.05 and 17.55%, with an average of 10.79%. The total phenotypic variance explained (TPVE) by all QTL under optimum conditions was 16.68% for population 1, 39.17% for population 2, and 9.32% for population 5. QTL for LNM were found only in population 2, and the TPVE was 11.50%.

The TPVE under LNO was 11.50% for population 2, 23.34% for population 3, 22.30% for population 4, and 30.54% for population 5. The average QTL effect size under optimum (0.14 t ha^−1^) conditions was the highest compared to LNM (0.04 t ha^−1^) or LNO (0.08 t ha^−1^) conditions. Interestingly, the favorable alleles of the QTL detected under all management conditions were contributed by both low-N tolerant and susceptible parents.

Forty-three significant QTL were identified for AD under optimum and low-N stressed conditions (Table 4 and Appendix A) across all chromosomes and populations. The number of QTL identified was 18 under optimum, 13 under LNM, and 12 under LNO conditions. The largest number of QTL was detected in population 2 (16), followed by population 1 (15). The phenotypic variance explained by each QTL ranged between 3.19% and 95.81%, with an average of 17.40%. The total proportion of phenotypic variance explained by all QTL under optimum conditions was 71.31% for population 1, 46.88% for population 2, 29.02% for population 3 (only one QTL), and 13.36% for population 4 (only one QTL). Under LNM, TPVE was 28.86% for population 1, 58.04% for population 2, 12.04% for population 3 (only one QTL), 8% for population 4 (only one QTL), and 37.71% for population 5. The TPVE under LNO was 47.27% for population 1, 33.58% for population 2, 46.11% for population 3, 25.45% for population 4, and 29.69% for population 5. The effect size of all QTL ranged from 0.11 to 3.56 days, with an average of 0.54 days. Despite many (24) individual QTL explaining more than 10% phenotypic variance under different management conditions, only one QTL with high effect size under all management conditions was found. This QTL was identified on chromosome 1 (343 cM) from population 3. The effect size of this QTL was 2.99 days under optimum, 2.23 days under LNM condition, and 3.56 days under LNO condition. ASI is another secondary trait related to flowering and indicates the tolerance of maize genotypes to low-N stress. Only three QTL, one in population 2, 4, and 5, which explained more than 10% of phenotypic variation for ASI (Table 4 and Appendix A) were identified. The TPVE by two QTL (39.41%) in population 5 was the highest attained in this study. The highest effect size for ASI was attained by these two QTL (0.53 and 0.32 days). Generally, the effect size for ASI varied between 0.05 and 0.53, days with an average of 0.14 days.

From the total of 25 QTL identified for PH from all populations on all chromosomes except chromosomes 4, 5, and 10, seven were under optimum, 10 under LNM, and eight under LNO conditions (Table 4 and Appendix A). Only population 2 had QTL for all three management conditions. Thirteen QTL from the three conditions individually explained more than 10% phenotypic variance for PH. For the QTL in population 2, the TPVE was 59.82% under optimum, 61.72% under LNM, and 49.52% under LNO conditions. For populations 1 and 3, the total phenotypic variance explained by all QTL under LNM were 20.40% (one QTL) and 44.52%, respectively. Three QTL together explained 26.45% of the phenotypic variation observed for PH in population 1. One QTL each in population 3, 4, and 5 explained 24.33%, 8.05%, and 13.39% of the observed phenotypic variance for PH. The effect size of the individual QTL for PH ranged from 0.87 to 8.34 cm, with an average of 2.19 cm. Like AD, a QTL on chromosome 1 (343 cM) of population 3 combined more than 10% phenotypic variance and the highest effect size for PH. Two other QTL on chromosomes 1 (194.94 to 195.75 Mbp) and 8 (92.20 to 94.58 Mbp) of population 2 explained high phenotypic variance (16.96% and 13.92%) and had high effect size (3.27 and 2.99 cm). Like PH, the largest number of QTL for EH and EPO (Table 4 and Appendix A) was identified from population 2. The QTL on chromosome 1 of population 3, which combines a higher proportion of phenotypic variance explained and high QTL effect for AD and PH, also had the same effect for EH.

Unlike other traits in this study, QTL for SEN were identified only from population 2, with the largest number being under LNO (Table 4 and Appendix A). The total phenotypic variance explained under optimum, LNM, and LNO conditions was 23.65%, 15.06%, and 45.87%, respectively. The highest amount of phenotypic variance and largest number of QTL under LNO indicates the genetic variability existing under LNO for SEN and the contrasting nature of the two parents that constituted population 2.

### 2.3. QTL Overlapping among Management Conditions for Each Trait

Discovering common QTL between different N conditions facilitates the identification of markers commonly used for optimum and low-N-stress breeding environments for a target trait. In this study, several common QTLs between optimum and N-stress conditions were identified for different traits, mainly in population 2 (Figure 2). For AD, one QTL (ADQTL1) under optimum conditions (282.41 to 283.19 Mbp) was overlapping with two QTL under LNM conditions (280.22 to 283.19 and 282.41 to 284.50 Mbp) on chromosome 1. Two common QTL (PHQTL1 and PHQTL2) were identified between optimum, LNM, and LNO for PH on chromosome 1 (from 194.94 to 195.75 Mbp and from 280.22 to 283.19 Mbp). In addition, one overlapping QTL (PHQTL3) was detected between LNM and LNO conditions on chromosome 1 (52.35 to 230.18 Mbp) of population 3 for PH. For EH, a common QTL (EHQTL1) was found between LNM and LNO conditions on chromosome 1 (194.94 to 195.75) in population 2. On chromosome 1 of population 3, two additional QTL (49.39 to 51.75 Mbp and 52.35 to 230.18 Mbp) named EHQTL2 and EHQTL3 were observed for EH between optimum and LNM conditions. One QTL (EPOQTL1) overlapping between optimum and LNM conditions for EPO was also found on chromosome 1 (57.29 to 58.36 Mbp) of population 2. In addition, closely linked QTL (EPOQTL2) were detected between optimum (191.54 to 191.75 Mbp) and LNO (193.15 to 193.75 Mbp) conditions, and overlapping QTL (EPOQTL3) were detected between optimum (33.66 to 36.72 Mbp) and LNM (34.34 to 34.55 Mbp) conditions on chromosome 4 of population 2. Other QTL correspondences (EPOQTL4 and EPOQTL5) under LNM and LNO conditions were found on chromosome 8 of population 1, and optimum and LNO conditions on chromosome 8 of population 2. For SEN, one QTL (SENQTL1) under optimum (203.94 to 204.47 Mbp) conditions was closely linked to a QTL (201.15 to 2012.14 Mbp) identified under LNM on chromosome 2 of population 2. No QTL correspondence was found among different management conditions for GY and ASI.

### 2.4. QTL for Multiple Traits in One/Different Population

Markers associated with common QTL among different traits and genetic backgrounds would facilitate the use of MAS to achieve yield improvement under low-N stressed conditions. Several multi-trait QTL were identified from all populations, except population 4. Because of large numbers of such QTL, only QTL with high PVE (≥10%) and relatively high effect size were reported here (Table 4 and Appendix A). In population 1, a QTL on chromosome 1 (45.04 to 47.83 Mbp) was involved in the control of AD, EH, and PH under LNM conditions. Another QTL in this population was found on chromosome 8 from 137.47 to 142.23 Mbp underlying AD under optimum and EH under LNM conditions. In population 2, four QTL on chromosome 1 were involved in the control of multiple traits: from 57.29 to 58.36 Mbp for EH under LNM and EPO under optimum and LNM; from 194.94 to 195.75 Mbp for EH under LNO and PH under optimum, LNM, and LNO; from 217.11 to 219.23 Mbp for GY and EH under optimum and from 280.22 to 283.19 Mbp for PH under LNO and AD under LNM conditions. A QTL stretch identified from population 3 on chromosome 1 (343 cM), spanning from 52.35 to 230.18 Mbp, was associated with AD under all the three N conditions and for PH under low-N conditions (LNM and LNO). A QTL on chromosome 8 (162.19 to 168.27 Mbp) of population 5 was involved in the control of GY, PH, and EH under LNO conditions. This is an adaptive QTL responsible for the control of GY, PH and EH only under severe low-N stress conditions.

Some QTL common between different genetic backgrounds were found for AD (chromosome 8 in population 1 and 2), and ASI (chromosome 3 of population 2 and 4). Both the upstream and downstream of the multi-trait QTL (57.84–69.87 Mbp) identified from population 3 also integrated QTL identified for GY and other secondary traits from other populations. This indicates that this specific region could be common among multiple genetic backgrounds and needs further research to fine-tune the position of the QTL responsible for the control of multiple traits and possible candidate genes involved.

## 3. Discussion

### 3.1. Yield Reduction, Variances, and Heritability

Yield reduction under low-N stress conditions is an indication of the role of N in the growth and development of maize. In this study, mean GY was reduced by 71% under severe N stress (LNO), and by 39% under moderate N stress (LNM) conditions. The yield reduction under moderate stress was similar to yield reduction reported previously [15] under low-N stress conditions. Under severe N stress, the yield reduction observed was similar to yield reduction reported in an earlier study [12] under severe low-N stress conditions during the wet season in Mexico. Many authors attributed yield reduction under low-N stress to reduction in number of kernels as a result of increased abortion [2,12,14,15]. A big gap between silk emergence and pollen shed (ASI) under low-N stress conditions is one of the causes for kernel abortion. The large difference in yield reduction under low N during main and off-seasons showed the seasonal variation of low-N environments during the rainy and dry seasons.

If GY under low-N stress is below 50% of the yield obtained under optimum N conditions, the yield reduction is related to mechanisms that impart tolerance to low-N stress [16]. In this study, GY obtained under severe stress conditions was only 29% of yield under optimum conditions and thus suitable for studying QTL underlying GY and secondary traits under low-N stressed conditions. However, since high levels of stress affect genetic variance and hence data [5] was collected for moderately N stressed environments also to capture QTL under all N-stress levels. In addition to GY, higher genetic variance was observed for some secondary traits under low-N conditions compared to optimum conditions, indicating the stress-adaptive nature of these traits [8], and therefore increases the power of QTL detection under low N [14].

### 3.2. QTL for GY and Secondary Traits under Optimum and Low-N Conditions

QTL underlying GY and secondary traits under optimum and low-N stress conditions could accelerate the development of NUE varieties. QTL analysis in this study identified 155 significant QTLs in five populations for seven traits under optimum, LNM, and LNO conditions. Some of these QTL were specific to only one trait, management condition, and population, while others were found across traits, management conditions, and populations. The distribution of the QTL also varied across the ten chromosomes of maize. The total number of QTL identified under the three conditions was comparable, indicating the existence of genetic variability under all three conditions. The highest number of QTL was detected for AD among traits, and in population 2 among populations. The result was consistent with the highest genetic variance observed for AD under all management conditions. The highest number of QTL in population 2 indicates the contrasting nature of the constituting parents for most traits under all management conditions. Chromosomes 1, 3 and 8 had the highest number of QTL and could be targeted for further QTL studies for GY and related secondary traits under both optimum and low-N stressed conditions. In another study [17], 13 QTL were identified under both low and high N levels. On chromosome 10, overlapping QTL were found for grain yield, ASI, and AD. Only three QTL were found to be common under low-N and optimal conditions, and this was for the “Stay Green” trait.

For any target trait, identifying common QTL across management conditions, traits, and populations is desirable for successful implementation of MAS schemes. GY is the primary trait of interest in most breeding programs in Sub-Saharan Africa, where maize is a staple food. Most breeding programs on the subcontinent develop new varieties under optimally managed experimental fields, and the resulting new varieties are commonly grown under N-limiting small-scale maize farms. The correlation between low- and high-N environments for grain yield has been reported to be low in both tropical [2,18] and temperate [19] environments. Lack of common QTL between low- and optimum-N conditions for GY in this study agrees with classical correlation studies, and shows distinct genetic mechanisms for GY under low and optimum N conditions. N-uptake efficiency under optimum and both uptake and utilization efficiencies under low-N conditions play a role in GY [15,20]. As such, GY improvement for low-N stressed environments should be through direct selection in target environments as previously suggested [2]. More QTL detected for GY under low-N than optimum conditions in this study indicates high genetic variability for GY under low-N stressed conditions. Most of these QTL explained more than 10% of phenotypic variance, suggesting that the markers associated with these QTL could be useful in MAS, to improve GY under low-N stressed conditions. MAS reduces the cost of extensive field testing and cuts the time required to develop NUE inbred lines and varieties [14] through a conventional plant-breeding approach. Previous QTL reports for GY under low and optimum N were highly variable [9,14]. Like the current study, no common QTL between optimum and low-N conditions for GY was found in a previous study [14]. Another study [12] reported QTL correspondence between optimum- and low-N conditions on chromosomes 1 and 3. Differences in number of markers, locations, and populations used in different studies could contribute to the different results.

Successful use of QTL for improving complex traits has apparently been hampered by their small effect size and lack of consistency across different genetic backgrounds [8] and locations. Identifying major effect QTL underlying single or multiple traits in various populations determines the successful use of QTL through MAS [8]. Since the advent of molecular markers, many QTL have been identified for GY and secondary traits under optimum and various stress environments, mainly based on individual or few mapping populations [4,5,12]. In this study, QTL were identified that were common between optimum and low-N stressed conditions (LNO or LNM) for all secondary traits except ASI. The QTL correspondence between optimum and low-N stressed conditions (LNM and LNO) for secondary traits was in agreement with high genetic and phenotypic correlation reported between optimum and low-N environments for each trait [2,18,19]. Common QTL for secondary traits justify the higher magnitude of correlation between optimum and low-N stress environments. The common QTL could be used to simultaneously improve each secondary traits for both optimum and low-N stressed conditions through markers associated with QTL identified under both optimum and low-N stressed conditions. However, QTL correspondences identified between optimum and low-N conditions for most secondary traits were not similar across populations, indicating the genetic-background-specific nature of QTL. In addition to common QTL for single traits between management conditions, multi-trait QTL would facilitate simultaneous improvement of traits or used in indirect selection for complex traits like GY through highly heritable and easily measurable traits. Populations 1, 2, 3, and 5 hosted QTL controlling multiple traits under different management conditions. The QTL identified in population 3 (chromosome 1: 52.35–230.18 Mbp; 343 cM), particularly, was remarkable, as it was common for AD, PH, and EH and integrated many QTL from other populations for AD, PH, GY, and ASI under optimum and low-N stressed conditions. QTL common between GY and easy-to-measure secondary traits can be used for indirect selection under low-N stress environments.

Recurrent selection with markers associated with GY QTL under both optimum and low-N stressed conditions can help to accumulate favorable alleles for GY. Finding major QTL for complex traits like GY is challenging and needs to consider other alternatives. One alternative approach is the use of indirect selection through QTL common between GY and secondary traits that are easy to measure, highly heritable, and controlled by a few genes compared to GY. An adaptive QTL to low-N stressed conditions identified on chromosome 8 (162.19 to 168.27 Mbp) in this study is promising for indirect selection for GY through selection for PH and EH. This QTL was identified from population 4 and was involved in the control of GY, PH, and EH under LNO conditions. The markers associated with this QTL can be used for simultaneous improvement of GY, PH and EH. A high correlation between GY and PH due to co-localized QTL for both traits was reported [12], and suggested inclusion of PH in selection indices as important trait for improving GY under low-N conditions. Pleiotropic QTL for GY with EPO and PH under low N was also reported previously [14], but on a different chromosome than seen in this study, indicating the possibility of identifying such QTL in different genomic regions across different genetic backgrounds.

## 4. Materials and Methods

### 4.1. Plant Materials

Five doubled haploid (DH) populations from the Improved Maize for African Soils (IMAS) and the Water Efficient Maize for Africa (WEMA) projects of CIMMYT were used in this study. The populations were developed through an in vivo DH technique [21]. Population 1 (CML494/CML550), population 2 (CML504/CML550), and population 3 (CML511/CML550) consisted of 108, 219, and 111 DH lines, respectively, developed from four inbred lines from CIMMYT heterotic group B. Population 4 (CML505/LaPostaSeqC7-F64-2-6-2-2-B-B) and population 5 (CML536/LaPostaSeqC7-F64-2-6-2-2-B-B) consisted of 159 and 109 DH lines, respectively, and were developed from three inbred lines from CIMMYT heterotic group A. CML550, LaPostaSeqC7-F64-2-6-2-2-B-B and CML494 are among the top 20 low-N donor lines identified from a 412 panel of lines tested under low N in multiple environments, while CML504, CML505, and CML536 were sensitive to low-N stress (data not shown). Consequently, one population used in this study represented low-N tolerant x tolerant (CML494/CML550), while the remaining four populations represented tolerant x sensitive crosses. The DH lines from the five populations were testcrossed to a tester from the complementary heterotic group. DH lines from population 1 were testcrossed to an inbred line tester CML312; population 2 and 3 were testcrossed to a single cross tester, CML312/CML443, and populations 4 and 5 were testcrossed to an inbred line tester, CML395. Testcross progenies from all five the populations were evaluated under optimum and managed low-N conditions in the main and off-seasons of 2014 and 2015 in Kenya and Rwanda. The low-N stress trials conducted during the off-season yielded significantly lower than the low-N stressed trials conducted during the main rainy seasons, and therefore separate genetic analyses were performed for N-stressed trials in the main rainy season (LNM) and off-season (LNO).

### 4.2. Field Experiments and Data Analysis

Testcross progenies from the DH lines derived from the five populations were arranged into five different trials and planted across one to ten sites in Kenya and Rwanda, between the main season (A) of 2014 and off-season (B) of 2015 (Table 5). In each trial, three to seven commercial checks were included. All optimum trials were evaluated during the main season. For low-N trials, some were evaluated during the main season, and others during the off-season, to capture the seasonal variability of N availability (Table 5). All trials were laid out in an alpha lattice design [22], each with two replications, except one site for each population 2 and population 3, which had three replications at the low-N site in Kiboko during the 2014 off-season. In all sites, plots were hand-planted with inter- and intra-row spacing of 0.75 and 0.25 m, except at Kiboko, where a row length of 4 m, with inter- and intra-row spacing of 0.75 and 0.2 m was used under both optimum and managed low-N stress sites. Two seeds per hill were planted in all sites. Three weeks after germination, plots were thinned to one plant per hill, to achieve a final plant density of 53,000 plants per hectare. At planting, only triple phosphate (46% P_2_O_5_) was applied to all low-N trials, at a rate of 50 kg of P_2_O_5_ ha^−1^. For optimum trials, diammonium phosphate (DAP)-fertilizer was applied at the recommended rate, for each location. Four weeks after planting, all optimum trials were top-dressed with urea fertilizer. The rate and type of fertilizer applied were the same during main and off-seasons.

Optimum and low-N trials at Kiboko were irrigated as required throughout the growing season, to avoid moisture stress, but trials on all other sites were rain fed. Except for N fertilization, the same management was applied to trials planted under optimum and low-N stress sites. The low-N trial fields were depleted of N for several seasons, and no N fertilizer was applied. Data were collected for GY, anthesis date (AD), anthesis-silking interval (ASI), plant height (PH), ear height (EH), ear position (EPO), and leaf senescence (SEN). GY was calculated from field weight by adjusting grain moisture to 12.5% and the shelling percentage to 80%. AD is the number of days from planting to when 50% of plants in the plot started shedding pollen on the main axis of the tassel. ASI was calculated as the difference between the number of days when 50% of plants in a plot emerged 2–3 cm silk and pollen shedding. PH and EH were measured in centimeters as a distance from the base of a plant to the first branch of the tassel and the upper most ear from ten representative plants, respectively. EPO was calculated as the ratio between PH and EH. SEN was recorded by visual assessment, using a 1–10 scale, where 1 indicates all leaves of all plants in a plot were green and 10 indicates that all leaves were dead. At harvest, edge plants were removed from all rows from trials planted under low N, to avoid border effects.

### 4.3. Genotyping, Genetic Maps, and QTL Analysis

Genomic DNA was extracted from young leaves collected in a bulk of 10 plants per entry, using a modified version of the CIMMYT high-throughput mini-prep Cetyl Trimethyl Ammonium Bromide (CTAB) method [23]. DNA samples were genotyped at the Institute of Biotechnology at Cornell University (http://www.biotech.cornell.edu/brc/genomics-facility), USA, using ApeKI as restriction enzyme and 96-plex multiplexing [24]. Genotyping by sequencing (GBS) data for a total of 955,120 SNP loci distributed across the ten maize chromosomes was received from the Institute of Genomic Diversity (IGD), Cornell University, USA. The genotype data was filtered with a minor allele frequency (MAF) of 0.05 and a minimum count of 95% of the sample size, using TASSEL v.5.2.24 software [25]. Only marker loci homozygous for both parents and polymorphic between the two parents were retained in all populations. Finally, markers were selected based on distance (more than 250 Mb apart) in order to get the number of markers handled by the QTL analysis software and to ensure uniform distribution of markers on the genome.

Linkage maps for all five populations were constructed using by QTL IciM mapping ver. 4.0.6.0. (http://www.isbreeding.net) software using a criterion of more than 3.0 logarithm of odds (LOD) [26]. Recombination frequencies between two linked loci were transformed into cM distances, using Kosambi′s mapping function [27]. QTL analysis was performed by using the across locations BLUPs for each population within each management condition. QTL associated with each trait were identified by using an inclusive interval mapping (ICIM) method implemented in the software QTL IciM Mapping v.4.0.6.0 [26]. The walking step in QTL scanning was 1 cM, and a LOD threshold of 3.0 was used to declare putative QTL [15]. The sign of the additive effects of each QTL was used to identify the direction (the origin of the favorable allele) and effect size of each QTL.

## 5. Conclusions

This study identified QTL underlying GY, AD, ASI, PH, EH, EPO, and SEN under optimum and low-N stressed conditions and SNP markers associated with each QTL. Some of the QTL identified were important to explain the genetic basis of correlation between optimum and low-N environments for GY and secondary traits. The genetic mechanism under optimum and low-N conditions seem distinct for GY, as there were no common QTL found under either condition. Generally, the cost of phenotypic evaluation under low-N environments is higher than under optimum conditions, due to the need for establishment and management of managed low-N stressed sites across regions. MAS through genomic regions associated with GY or indirectly through secondary traits correlated with GY under low-N environments would help to reduce the cost of breeding for stress environments. QTL explaining more than 10% phenotypic variance and relatively higher effect size can be used for fine mapping and/or marker-assisted breeding for rapid GY improvement under optimum and low-N stressed conditions.

## Figures and Tables

**Figure 1 ijms-21-00543-f001:**
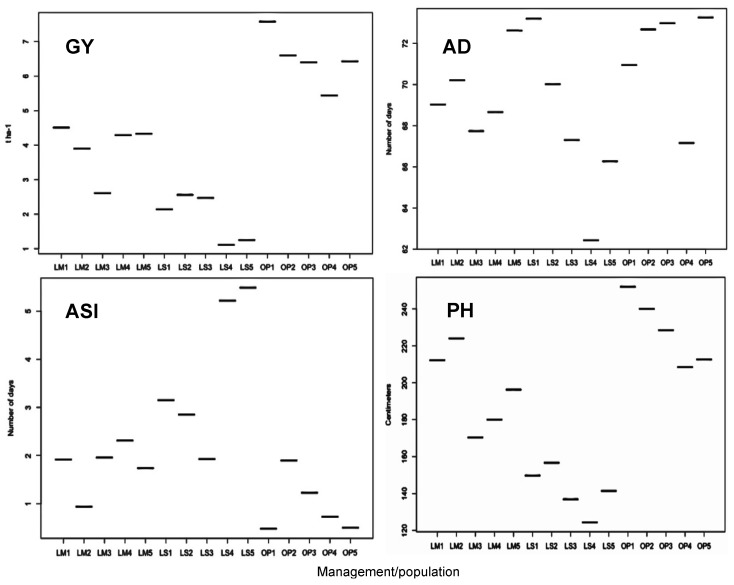
The mean of grain yield (GY), anthesis date (AD), anthesis-silking interval (ASI), and plant height (PH) under optimum (OP), moderately low-N stress (LM) and severely low-N stressed (LS) conditions. The numbers after the management conditions on *x* axis indicate populations 1 to 5.

**Figure 2 ijms-21-00543-f002:**
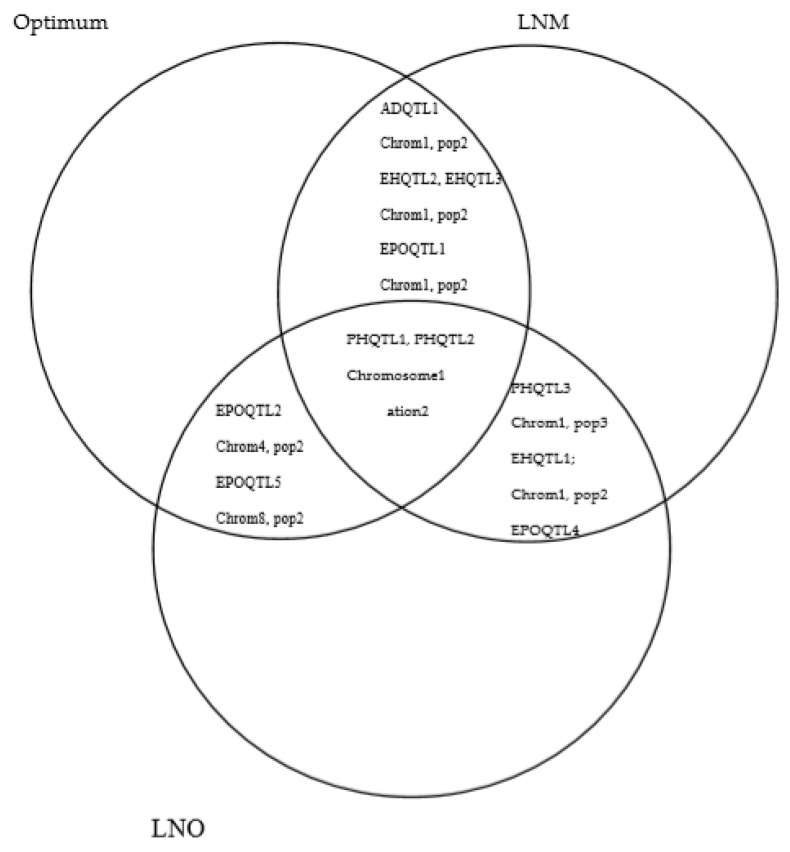
Overlapping QTLs between three field management conditions, and their chromosome locations in different populations.

**Table 1 ijms-21-00543-t001:** Number of markers and total map distance observed in each population used for QTL analysis.

Chr.	CML494/CML550		CML504/CML550		CML511/CML550		CML505/LaPostaSeqC7-F64-2-6-2-2-B-B		CML536/LaPostaSeqC7-F64-2-6-2-2-B-B	
	SNPs	Distance (cM)	SNPs	Distance (cM)	SNPs	Distance (cM)	SNPs	Distance (cM)	SNPs	Distance (cM)
1	371	644.9	404	511.5	285	642.5	314	1068.4	311	512.5
2	242	432.4	310	460.6	237	428.5	216	841.4	256	270.9
3	233	417.7	286	538.8	197	434.7	259	961.6	237	489
4	238	359.3	325	336.7	244	470.7	254	796.0	277	381.5
5	211	502.7	283	539.4	193	568.4	118	581.7	211	311.1
6	138	249.6	195	320.3	180	225.2	172	808.5	177	312.4
7	172	420.8	228	375.4	153	318.4	151	671.1	152	432.7
8	195	379.9	250	441.6	177	322.5	170	506.0	176	247.9
9	162	103.9	221	294.7	139	199.5	162	543.7	140	236
10	142	177.1	197	185.6	157	261.5	169	414.7	149	232.4
Total	2104	3688.3	2699	4004.6	1962	3871.9	1985	7193.1	2086	3426.4

**Table 2 ijms-21-00543-t002:** Number of QTL detected for grain yield (GY), anthesis date (AD), anthesis-silking interval (ASI), plant height (PH), ear height (EH), ear position (EPO), and leaf senescence (SEN) under optimum and low-nitrogen stress in main rainy season (LNM) and off-season (LNO), across the ten chromosomes.

Chr.	LNM								LNO								OPT								Total
	AD	ASI	EH	EPO	GY	PH	SEN	Tot	AD	ASI	EH	EPO	GY	PH	SEN	Tot	AD	ASI	EH	EPO	GY	PH	SEN	Total	
1	5	1	4	1		4	1	16	7	1	3	4	2	3		20	3		5	2	2	3		15	51
2	2						1	3					1			1				1	1	1	1	4	8
3	1	1	3			2		7		3			3	2	2	10	3	2	1	1		1	1	9	26
4			1	2				3	1			1	1			3	3			2				5	11
5			2					2	1						1	2	3	1	1	1				6	10
6	1					1		2				1			1	2	1		1	1			1	4	8
7	1		2		1	1		5	1							1	1	1						2	8
8			2	1		2		5	1	1	2	2	1	2		9	4			1		1		6	20
9	1							1	1					1	1	3		1				1		2	6
10	2	1	1		1			5											2					2	7
Total	13	3	15	4	2	10	2	49	12	5	5	8	8	8	5	51	18	5	10	9	3	7	3	55	155

**Table 3 ijms-21-00543-t003:** Genetic characteristics of detected QTLs for grain yield (GY) under optimum and low-nitrogen stress in main season (LNM) and off-season (LNO) in DH lines derived from five bi-parental populations.

Population	Management	Chr.	Pos (cM)	Left Marker	Right Marker	LOD	PVE (%)	TPVE (%)	Add	Fav Allele
CML550 × CML494	OPT	2	318	S2_15120146	S2_15909091	4.48	17.23	16.68	0.12	CML550
CML550 × CML504	OPT	1	46	S1_283186611	S1_280222332	4.43	6.31	39.17	−0.12	CML504
	OPT	1	183	S1_219232023	S1_217114738	10.24	15.04		−0.18	CML504
	LNM	7	278	S7_106325823	S7_105221050	3.39	6.84	11.50	−0.04	CML504
	LNM	10	143	S10_11189892	S10_10138689	3.06	6.05		0.04	CML550
	LNO	2	173	S2_200745112	S2_202138908	3.10	6.66	11.55	−0.09	CML504
CML550 × CML511	LNO	3	228	S3_127100888	S3_128233351	4.69	17.55	23.34	−0.10	CML511
	LNO	4	177	S4_165623425	S4_162255436	3.71	13.56		−0.09	CML511
CML505/LaPostaSeqC7-F64-2-6-2-2-B-B	LNO	1	531	S1_177877619	S1_183811363	3.91	10.14	22.30	0.06	LP
	LNO	3	169	S3_204998702	S3_203869859	4.38	10.34		−0.05	CML505
CML536/LaPostaSeqC7-F64-2-6-2-2-B-B	LNO	1	439	S1_41220359	S1_39739703	3.64	9.90	38.54	0.08	LP
	LNO	3	345	S3_38439419	S3_31449087	3.68	10.10		0.08	LP
	LNO	8	157	S8_166372615	S8_168274395	3.77	10.54		0.08	LP

LOD, logarithm of odds, PVE, phenotypic variance explained, TPVE, total phenotypic variance explained.

**Table 4 ijms-21-00543-t004:** Summary of detected QTL for measured characteristics under optimum and low-nitrogen stress in main season (LNM) and off-season (LNO) in DH lines derived from five bi-parental populations.

Characteristic	Population	Management	Number of QTL	TPVE (%)
	CML550/CML494	OPT	8	71.31
Anthesis date		LNM	2	28.86
		LNO	4	47.27
	CML550/CML504	OPT	8	46.88
		LNM	6	58.04
		LNO	2	33.58
	CML550/CML511	OPT	1	29.02
		LNM	1	12.04
		LNO	2	46.11
	CML505/LaPostaSeqC7-F64-2-6-2-2-B-B	OPT	1	13.36
		LNM	1	8.00
		LNO	2	25.45
	CML536/LaPostaSeqC7-F64-2-6-2-2-B-B	LNM	1	37.71
		LNO	1	29.69
Anthesis- silking	CML550/CML504	OPT	4	31.26
interval		LNM	3	24.10
		LNO	1	12.84
	CML505/LaPostaSeqC7-	OPT	3	11.70
	F64-2-6-2-2-B-B	LNO	2	19.68
	CML536/LaPostaSeqC7- F64-2-6-2-2-B-B	LNO	2	39.41
Plant height	CML550/CML494	LNM	1	20.40
		LNO	2	26.45
	CML550/CML504	OPT	7	59.82
		LNM	6	61.72
		LNO	3	49.52
	CML550/CML511	LNM	3	44.52
		LNO	1	24.33
	CML505/LaPostaSeqC7-F64-2-6-2-2-B-B	LNO	1	8.05
	CML536/LaPostaSeqC7-F64-2-6-2-2-B-B	LNO	1	13.39
Ear height	CML550/CML494	LNM	5	53.16
		LNO	1	24.29
	CML550/CML504	OPT	7	56.64
		LNM	6	52.26
		LNO	3	27.42
	CML550/CML511	OPT	2	22.87
		LNM	2	23.70
	CML505/LaPostaSeqC7-F64-2-6-2-2-B-B	OPT	1	14.79
		LNM	1	
	CML536/LaPostaSeqC7-F64-2-6-2-2-B-B	LNM	1	23.10
		LNO	1	11.40
Ear position	CML550/CML494	OPT	1	10.53
		LNM	1	23.30
		LNO	1	28.75
	CML550/CML504	OPT	5	38.35
		LNM	3	33.54
		LNO	5	38.34
	CML550/CML511	OPT	2	28.05
	CML505/LaPostaSeqC7-F64-2-6-2-2-B-B	OPT	1	15.30
	CML536/LaPostaSeqC7-F64-2-6-2-2-B-B	LNO	2	31.04
Senescence	CML550/CML504	OPT	3	23.65
		LNM	2	15.06
		LNO	5	45.87

TPVE, total phenotypic variance explained.

**Table 5 ijms-21-00543-t005:** Trial management information for testcross progenies of five double haploid bi-parental populations evaluated under optimum and low-N conditions in Kenya and Rwanda, from seasons 2014A to 2015B.

Trial Name	Site	Year *	Nr of Entries	Nr of Checks	Management	Rep	Mean GY
**CML494/CML550; Tester: CML312**							
**Population Size: 108**							
15B-EMB-8	Embu	2015B	110	2	LNO	2	2.24
15A-KKM-9	Kakamega	2015A	110	2	Opt	2	7.13
15A-RWA-3	Rwanda	2015A	110	2	Opt	2	6.72
15A-KBK-1	Kiboko	2015A	110	2	Opt	2	8.88
15A-KBK-2	Kiboko	2015A	110	2	LNM	2	4.00
15B-KBK-5	Kiboko	2015B	110	2	LNO	2	2.05
15A-KIT-7	Kitale	2015A	110	2	LNM	2	5.43
15A-MTW-4	Mtwapa	2015A	110	2	LNM	2	4.10
**CML504/CML550; Tester: CML312/CML443**							
**Population Size: 219**							
14A-ALU-9	Alupe	2014A	224	5	LNM	2	3.60
14A-EMB-5	Embu	2014A	224	5	LNM	2	4.22
14A-KKM-3	Kakamega	2014A	224	5	LNO	2	2.40
14A-KKM-4	Kakamega	2014A	224	5	Opt	2	5.35
14B-KBK-1	Kiboko	2014B	224	5	LNO	3	2.92
14B-KBK-2	Kiboko	2014B	224	5	LNO	3	2.39
14A-KBK-1	Kiboko	2014A	224	5	Opt	2	6.49
14A-KBK-2	Kiboko	2014A	224	5	Opt	2	10.60
14A-KIT-10	Kitale	2014A	224	5	Opt	2	5.67
14A-KIT-8	Kitale	2014A	224	5	Opt	2	4.91
**CML511/CML550; Tester: CML312/CML443**							
**Population Size: 111**							
14A-ALU-9	Alupe	2014A	116	6	LNM	2	2.65
14A-KKM-3	Kakamega	2014A	116	6	LNM	2	2.72
14A-KKM-4	Kakamega	2014A	116	6	Opt	2	4.99
14B-KBK-1	Kiboko	2014B	116	6	LNO	3	2.12
14B-KBK-2	Kiboko	2014B	116	6	LNO	3	2.83
14A-KBK-2	Kiboko	2014A	116	6	Opt	2	10.40
14A-KBK-5	Kiboko	2014A	116	6	Opt	2	6.07
14A-KIT-10	Kitale	2014A	116	6	Opt	2	5.19
14A-KIT-8	Kitale	2014A	116	6	Opt	2	5.35
14A-MTW-6	Mtwapa	2014A	116	6	LNM	2	2.47
**CML505/LaPostaSeqC7-F64-2-6-2-2-B-B; Tester: CML395/CML444**							
**Population size: 159**							
WET15A-EVALITC-08-1	Kakamega	2015A	174	6	Opt	2	7.53
WET15A-EVALITC-08-2	Kiboko	2015A	174	6	Opt	2	5.81
WET15A-EVALITC-08-5	Kiboko_LN	2015A	174	6	LNM	2	4.29
WET15A-EVALITC-08-6	Kiboko2_LN	2015A	174	6	Opt	2	5.12
WET15A-EVALITC-08-8	Kiboko3_LN	2015B	174	6	LNO	2	1.11
**CML536/LaPostaSeqC7-F64-2-6-2-2-B-B; Tester: CML395/CML444**							
**Population Size: 109**							
WET15A-EVALITC-11-1	Kakamega	2015A	130	8	Opt	2	8.58
WET15A-EVALITC-11-2	Kiboko	2015A	130	8	Opt	2	5.00
WET15A-EVALITC-11-5	Kiboko_LN	2015A	130	8	LNM	2	4.33
WET15A-EVALITC-11-6	Kiboko2_LN	2015A	130	8	Opt	2	5.71
WET15A-EVALITC-11-8	Kiboko3_LN	2015B	130	8	LNO	2	1.25

* A, main season; B, off-season; LN, low N; opt, optimum N; LNM, low-N site during main season; LNO, low N during off-season (severe low-N stress); Nr., number; Mgt, management; Rep, replication; σ^2^g, genotypic variance; σ^2^e, error variance; h^2^, broad sense heritability.

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
