# Peer review of "Genetic Dissection of Grain Yield and Agronomic Traits in Maize under Optimum and Low-Nitrogen Stressed Environments"

_ijms, 2020, doi:10.3390/ijms21020543_

Round 1
Reviewer 1 Report
The authors present a workmanlike study of QTL in maize under N2 stress using GBS. I have no major issues with the work and recommend it for publication, but there are several quite dry and technical data tables that would probably be better summarised and moved to supplementary data.
Author Response
Comments
The authors present a workmanlike study of QTL in maize under N2 stress using GBS. I have no major issues with the work and recommend it for publication, but there are several quite dry and technical data tables that would probably be better summarised and moved to supplementary data.
Answer: Thanks for the constructive comment. We agree with your comments on lengthy tables. We have made one table (Table 5) which summarizes what was Tables 5-10. These tables were then made supplementary tables.
Reviewer 2 Report
This manuscript describes results of genetic dissection of several maize traits under optimum and low nitrogen stressed environments. From the perspective of bioinformatics, the basic procedure and results are technically sound. Major comments: 1. This manuscript lacks further exploration of the QTL found in the studies. Finding the QTL through established methods and materials are easy, it could be better to perform some functional analysis on these QTL regions, such as gene annotation and cross-validation through other studies. 2. As the study mentioned high density data obtained already, GWAS analysis is much more appreciated for it increase the resolution than the region of QTL Minor comment: 1. Figure 1 is totally messed in the PDF version downloaded from the system
Author Response
Reviewer 2
This manuscript describes results of genetic dissection of several maize traits under optimum and low nitrogen stressed environments. From the perspective of bioinformatics, the basic procedure and results are technically sound.
Major comments: 1. This manuscript lacks further exploration of the QTL found in the studies. Finding the QTL through established methods and materials are easy, it could be better to perform some functional analysis on these QTL regions, such as gene annotation and cross-validation through other studies.
We agree with your suggestion. The current study was divided into three parts in first objective is to discover the QTL in biparental populations which is presented in this manuscript. Regarding connecting these QTL with candidate genes, we feel, because of large QTL confidence intervals, may not be precise in relating to candidate genes. In our next manuscript we are planning to reduce this CI through across biparental populations mapping or through joint linkage association mapping and GWAS study where we probably reduce the big confidence intervals of the QTLs identified in the current study and get specific regions then we are planning to connect with candidate genes and also planning to explore potential of genomic selection for NUE improvement. Since we had 5 populations evaluated in optimum and low N management, data is too much to include everything in one MS, so our next manuscript on GWAS will cover the most of questions related to candidate genes. Functional analyses is another interesting area of research, in our applied research we are limiting our research to till get the trait linked SNPs and their utility but it will be an interesting study to consider in future research
Comment 2.
As the study mentioned high density data obtained already, GWAS analysis is much more appreciated for it increase the resolution than the region of QTL
Answer
Thanks for your comment. Yes we are in the stage to submit GWAS results based manuscript to the same journal soon.
Minor comment: 1.
Figure 1 is totally messed in the PDF version downloaded from the system
Answer: We modified the figure in the revised manuscript
Round 2
Reviewer 2 Report
QTLs are still important and essential for maize study, but interpreting is vital in this area which is full of annotations on high-density SNPs from resequencing. According to the authors' responses, they deliberately separate the full studies, it decreases the value of the manuscript significantly and unacceptable to me.
I insist to see annotations, at least part of them in these studies, to make this manuscript a paper.